# Umbrella review of international evidence for the effectiveness of school-based physical activity interventions

**Abdullah Alalawi**[1,2]*, **Lindsay Blank**[1], **Elizabeth Goyder**[1]

**1** Health and Related Research, University of Sheffield, Sheffield, United Kingdom, **2** Al Qunfudah Health Sciences College, Umm Al-Qura University, Makkah, Saudi Arabia

* aoalalawi1@sheffield.ac.uk

## Abstract

### Background

Obesity and physical inactivity among children and young people are public health concerns. Despite the wide variety of interventions available to promote physical activity, little is known about which interventions are most effective. This review aimed to evaluate the existing literature on school-based interventions that aim to increase physical activity among children and young people aged 6 to 18 years.

### Methods

A systematic review of reviews was undertaken. We searched for systematic reviews and meta-analyses published between December 2017 and January 2024 using databases such as PubMed, Scopus, and CINAHL. Titles and abstracts were independently screened by two reviewers, who also conducted data extraction and quality assessments. We focused on outcomes like changes in physical activity levels and body mass index to assess the effectiveness of the interventions.

### Results

A total of 23 reviews examining school-based physical activity interventions met the inclusion criteria, comprising 15 systematic reviews and 8 meta-analyses. All reviews (N = 23) were implemented in the school setting: three in primary schools, seven in secondary schools, and thirteen targeted both primary and secondary schools. The findings demonstrated that six reviews reported a statistical increase in physical activity levels among the target population, and one review found a decrease in body mass index. The most promising interventions focused on physical activity included within the school curriculum and were characterised as long-term interventions. 20 out of 23 reviews assessed the quality of primary studies.

### Conclusion

Some interventions were promising in promoting physical activity among school-aged children and young people such as Daily Mile, Active Break, and Active transport while multi-

**Data Availability Statement:** All relevant data are within the manuscript and its Supporting Information files.

**Funding:** The author(s) received no specific funding for this work.

**Competing interests:** The authors have declared that no competing interests exist.

component interventions seem to be positively effective in reducing BMI. Future efforts should focus on long-term, theory-driven programmes to ensure sustainable increases in physical activity.

## Introduction

The World Health Organization (WHO) identifies childhood obesity as a significant public health concern in the 21st century. In 2016, a staggering 41 million children below the age of five were reported to be overweight, predominantly residing in developing nations [1]. This issue is closely linked to sedentary behaviors, which have become increasingly prevalent; currently, it is estimated that up to 81% of adolescents worldwide are not meeting the recommended levels of 60 minutes of moderate to vigorous physical activity (MVPA) daily [2]. Such physical inactivity is a major contributor to the rise in non-communicable diseases, such as diabetes, cardiovascular diseases, and metabolic syndrome in later life [3, 4].

The adverse effects of childhood obesity extend beyond physical health, impacting psychological aspects as well; children facing obesity issues often suffer from poor self-image and low self-esteem, elevating their risk of developing eating disorders and leading to a diminished quality of life [5]. Contributing to the complexity of this challenge are significant societal and lifestyle changes observed over recent decades in Europe, characterized by a shift towards high consumption of unhealthy fats and added sugars and a marked decrease in physical activities [6, 7].

Numerous programmes have been developed to promote healthy lifestyles and combat childhood obesity, with schools commonly the setting for such interventions. This strategic choice leverages the school's capacity to reach children across diverse ethnic and socio-economic backgrounds, ensuring that health-promoting efforts are broadly accessible [8, 9]. Schools not only provide a structured environment for implementing physical activity programmes but also inherently promote physical activity through compulsory sports education and encouraging active commuting among students [10–12]. Despite these widespread initiatives, there remains a significant variation in the success rates of these programmes, often due to differences in programme design, implementation fidelity, cultural adaptations, and other contextual factors.

A previous umbrella systematic review [9] synthesized evidence on interventions to increase physical activity among children and young people across various settings, including school-based programmes. It found that interventions involving physical activity embedded within the school curriculum, long-term initiatives, initiatives with teacher involvement, and family support show the most promise in enhancing physical activity levels. The review includes studies conducted in school, preschool, family, and community settings, highlighting the comprehensive scope of its analysis and the pivotal role of schools in promoting physical activity.

This umbrella systematic review aims to bridge an evidence gap by updating and providing a comprehensive overview that includes more recent studies on the effectiveness of school-based physical activity interventions to promote physical activity among children, and young people aged 6 to 18 years old worldwide.

Methods

An umbrella systematic review incorporating literature search and narrative synthesis was conducted to identify, evaluate, and synthesize relevant evidence from intervention studies

conducted in school settings globally. This review was conducted in accordance with the PRISMA guidelines for systematic reviews (S1 Checklist). The review protocol was registered with the PROSPERO International Prospective Register of Systematic Reviews, under registration number CRD42023494953.

## Formulation of research question

The PICO framework was considered to formulate the aim of this research by describing the population (P), intervention (I), comparison (C) and outcome (O), as summarised in Table 1 below. The value of the PICO framework is that it breaks down complex questions into simpler phrases that also form the inclusion and exclusion criteria (Fenton & Baxter, 2016).

## Inclusion and exclusion criteria

Ideally, including a large body of evidence is desirable because it provides more material from which to derive decisions. However, given the extensive volume and diversity of research on school-based physical activity interventions worldwide, we decided to conduct an umbrella review of relevant systematic reviews, which included primary studies on the effectiveness of these interventions. Consequently, we applied the following inclusion and exclusion criteria to select the most appropriate articles for inclusion in this umbrella review.

## Selection criteria

This umbrella review included systematic reviews and meta-analyses that incorporated various study designs, including randomised controlled trials (RCTs), cluster-randomised controlled trials (c-RCT), controlled trials (CT), quasi-experimental studies, and observational studies. Excluded from consideration were studies that did not adopt a systematic review approach, narrative reviews, and primary studies. The reviews were deemed eligible if they involved participants between the ages of 6 to 18 years. Included studies evaluated at least one intervention for increasing physical activity among the targeted population, were delivered in school and included quantitative evaluations of PA. Reviews published before 2018 were excluded because the previous related umbrella review was based on a search for studies published from January 2010 until November 2017 [9]. Studies published in Arabic or English language were included.

## Literature search process

Systematic searches were conducted in PubMed/MEDLINE, Scopus and CINAHL. These databases were chosen not only because they are easily accessible but also because they contain a large volume of systematic reviews and meta-analysis, due to their public health scope. Furthermore, we searched reference lists in included reviews to identify relevant studies. Details of the search strategy are outlined in the (S1 Appendix in S1 File).

**Table 1. PICO framework.**

| Population | School children and young people aged 6–18 years. |
|---|---|
| Intervention | School-based PA as either a primary intervention or a component of a multi behavioural intervention |
| Comparison | No intervention or usual school provision of PA, comparison of two or more interventions. |
| Outcome | The primary outcome is the self-reported or objectively measured change in PA levels. Secondary outcome includes changes in Body Mass Index. |

### Studies selection

All records were exported to Mendeley reference manager software. Duplicate records and all studies whose titles or abstracts did not meet the inclusion criteria were excluded. Screening of studies was conducted by two independent reviewers and, titles and abstracts of records retrieved from searches were screened for inclusion, and any disagreement was resolved by discussion with a third reviewer.

### Data extraction

A standardised approach was used to extract data from the selected studies [13], including study details such as the author(s), publication year, title, design, the number of studies included in the reviews, descriptions of interventions, duration of interventions and follow-ups, and outcomes measured, including physical activity levels and BMI, along with the main findings.

### Quality assessment

To evaluate the methodological quality and assess the risk of bias within the included reviews, the updated AMSTAR 2 version was utilised for analysing systematic reviews and meta-analyses, adhering to the recommendations specified by [14]. This tool enabled the systematic reviews to be classified based on their overall quality into four categories: high, moderate, low, or critically low. any disagreement was resolved by discussion with a third reviewer.

## Results

This results section contains three main parts: the search outcome, data extraction and synthesis, and quality of included reviews.

### Search outcome

The search was conducted in January 2024. The initial literature search produced a total of 2365 studies: PubMed/MEDLINE 1359, Scopus 290 and CINAHL 716. There were 682 duplicates, and after their exclusion, the 1683 remaining records' titles and abstracts were screened against the inclusion and exclusion criteria. The full texts of the remaining 60 studies were screened which led to the exclusion of 35 studies. The main reasons for exclusion were (1) having an inappropriate target population (n = 8); (2) being protocols (n = 2); (3) not school-based (n = 5); (4) measured different outcomes (n = 17) and (5) did not adopt a systematic review approach (n = 3). Further articles (n = 4) were identified from reference lists of included reviews. Details on excluded studies can be found in (S2 Appendix in S1 File). A PRISMA flow diagram shows search results (Fig 1).

### Review characteristics

A total of 23 studies met the inclusion criteria, comprising 14 systematic reviews, eight meta-analyses, and one scoping review that employed a systematic review methodology. The characteristics of these reviews are detailed in (S3 Appendix in S1 File)). The majority of the reviews were international in scope, with only four reviews targeting specific regions: one focused exclusively on Asian countries [15], another on the Middle East and Arabic-speaking countries [16], a third on school children in Latin America [17], and a fourth on adolescents in Australia [18]. All reviews included both males and females except one review which targeted only girls [19].

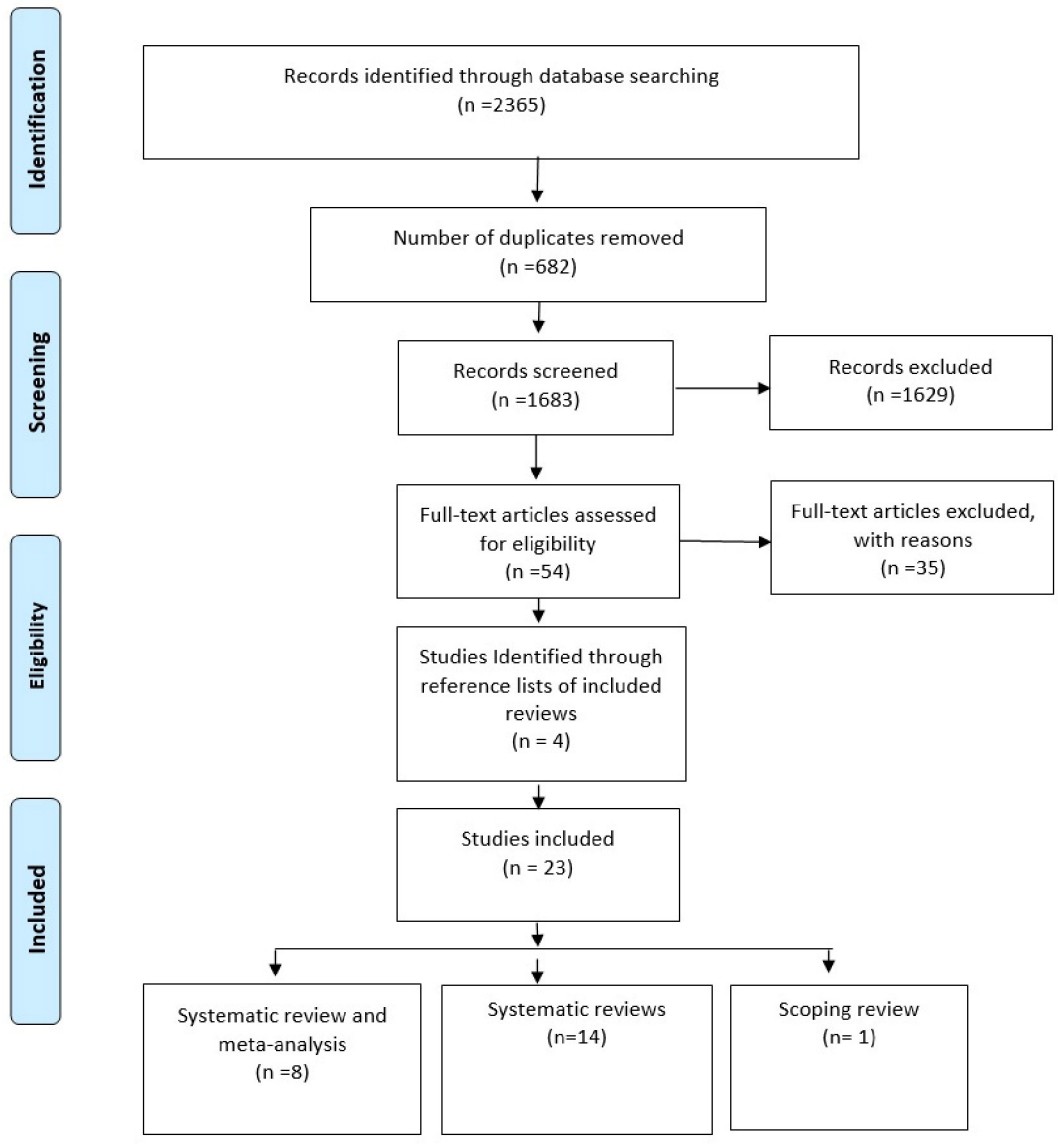

**Fig 1. PRISMA flow diagram.**

### School-based interventions

Twenty-three studies were included in this umbrella review, and these studies were grouped according to age and intervention type.

### Interventions for children aged 6 to 12 Years

Seven reviews [20–26] concerned interventions conducted in the primary schools setting for children aged between 6 and 12 years.

**Daily Mile programme.**   Two reviews [20, 23] examined the effectiveness of this intervention, Breslin and their colleagues observed significant variability in outcomes across studies. Specifically, Breslin et al. reported one intervention session led to a significant increase in MVPA, with the intervention group achieving 10.67±2.74 minutes of MVPA compared to 0.44

±0.95 minutes in the control group. Another study within the same review [23] noted a significant increase in PA levels after eight months, showing a daily increase of 9.1 minutes of MVPA in the intervention group. Regarding physical fitness, six primary studies in Breslin's review documented a notable improvement in the intervention group versus the control group, although comparing results was challenging due to the varied fitness tests used.

Overall, Breslin and colleagues concluded that sustained participation in the Daily Mile leads to increased MVPA and physical fitness among school-aged children, without significant impacts on academic performance or BMI. Only one study highlighted a short-term enhancement in mental health [23]. Similar results were reported by [20]. However, they also reported some implementation barriers for "The Daily Mile programme" such as repetitive nature, time constraints associated with competing curriculum demands, and inadequate facilities regularly necessitate the adaptation and development of the original Daily Mile intervention format by schools and teachers [20]. It should be noted that Breslin et al review and Hanna et al review included largely the same primary studies and the quality assessment for included studies conducted by Breslin et al revealed that one study was excellent, six were good, five were fair and none of them was poor quality.

**Active-break intervention.** Two systematic reviews [21, 25], evaluated the effectiveness of this intervention, Carrasco-Uribarren et al, found a significant improvement in both PA and MVPA, levels. Six included studies in [25] review, recorded a significant increase in in-school PA (SMD = 0.46; 95% CI: 0.28, 0.64; I2: 52%). Furthermore, five studies recorded a significant increase in MVPA (MD = 3.20; 95% CI: 3.06, 3.35; $I^2$: 0%), however, the reduction was not significant for sedentary behavior (SMD = −0.95; 95% CI: −2.06, 0.15; $I^2$: 98%). Masini and their colleagues also reported similar findings as they reported that Active-break interventions had a significant effect in increasing PA levels, MVPA and step count among primary school children and their meta-analysis demonstrated a statistically significant result for the step count (p < 0.00001, CI 95% −0.71,1.21) (random model I 2 = 0%) [21]. It should be noted that both reviews [21, 25], included five matching primary studies and the methodological quality of most of the included studies in the review of [25], was generally low and the risk of bias was very high. Furthermore, five RCTs in the review conducted by Masini and colleagues were poor quality, and 11 cohort studies were medium quality.

**Theory-based interventions.** One review [24], investigated universal interventions (n = 178), 62 interventions were theory based and the duration on average was 44 weeks (Range: 1–410 weeks). 146 studies assessed the PA outcomes as 74 studies used accelerometers, 25 studies used pedometers, 58 used questionnaires and 8 studies used direct observation. 141 included studies in this review evaluated the effectiveness of the intervention on PA outcomes post-intervention, and they recorded at least one significant intervention effect on measured PA, while 48 included studies did not show any significant increase in PA. In terms of long-term effectiveness, nine out of nineteen studies showed at least one significant intervention effect on measured PA, while ten studies did not indicate any intervention effect on the PA. In brief, the review of [24] found a high percentage of effective interventions, some single-feature interventions were found to be effective while no specific feature combination seemed to be associated with better intervention effectiveness. Therefore, theory-based single as well as multi-feature interventions seem to have the potential to improve effectiveness concerning PA, CRF, and sedentary behaviour [24]. It should be noted that the quality of the included studies in this review was unknown since the authors did not assess the risk of bias.

**Multi-component interventions.** In the review of [26] most of their interventions (70%) were multi-component interventions and they found that interventions in children resulted in a small positive treatment effect in the intervention group compared to the control group (2.14; 95% CI = 0.77, 3.50). There was no significant effect on sedentary behaviors, energy

intake, and fruit and vegetable intake but significant reductions were found between groups in BMI kg/m2 (−0.39; 95% CI = −0.47, −0.30) and BMI z-score (−0.05; 95% CI = −0.08, −0.02) in favour of the intervention. However, the quality of most of the included studies in this review was at low risk of bias.

**Physical education lessons.** The last review of this age group [22] was to examine interventions to increase MVPA content in physical education. Their intervention "Born to Move physical activity and fitness intervention alongside a regular PE lesson" found favorable intervention effects on children's MVPA in all the studies, with a pooled effect of 14.3% higher lesson time in MVPA in the intervention groups, equivalent to around a 9-minute improvement in MVPA per 1 hour of PE lesson time. The range was from a 4% to 30% difference in the MVPA content of PE lessons. Evidence quality was generally high in this review.

## Interventions for adolescents aged 12 to 18 Years

Out of the twenty-three reviews, only three reviews [18, 19, 27] investigated interventions conducted in secondary schools for adolescents aged between 12 and 18 years.

**Multi-component interventions.** Buru and their colleagues reviewed three types of interventions: physical activity, sedentary behavior and nutrition. PA was used in 11 out of 13 reviewed studies, the shortest intervention was four months and the longest was two years with a follow-up of three years. They found that only two of the included studies reported a significant increase in active transport, in terms of change in PA. Furthermore, only four of the included studies reported a significant decrease in BMI, whereas eight studies did not find any significant changes in BMI. Furthermore, three included studies recorded a significant decrease in body fat and only one study reported a significant decrease in body weight for the intervention group as compared to the control group. Therefore, Buru and their colleagues thought that despite the weak evidence of intervention efficacy for most of the reviewed studies, school-based interventions with multi-component combinations of physical activity, nutrition, and alignment to a theory yielded promising results. The quality assessment of included studies revealed that 11 studies were of medium quality and two studies were of high quality.

In the systematic review that was focused on only adolescent girls [19], most of their interventions were multi-component, four interventions were modified PE lessons and two were educational based, the shortest intervention was six months and the longest was three years. 13 interventions were delivered by the school staff, two were delivered by instructors and two by research team. Ten included studies used accelerometers to measure PA, nine studies used self-report questionnaires and two studies combined self-report and accelerometers. Owen with their colleagues found that school-based interventions provided a small, but significant positive effect on PA levels among adolescent girls (k = 16, g = 0.07, p = 0.05). The quality assessment for included studies revealed that ten studies were classified as having a moderate risk of bias, six studies were weak, three were strong and one had a very strong risk of bias.

**Active breaks intervention.** The systematic review conducted by [27], aimed to examine the impact of school-based physical activity interventions of "active breaks" on PA levels and other outcomes such as classroom behaviour, cognitive functions and well-being. They reviewed three studies about physically active lessons and active break interventions. Intervention periods varied from 11 weeks to 7 months. They found that two of the included studies demonstrated a positive effect of active breaks on students' classroom behavior and quality of life whereas, one study reported a positive effect in the increase in school physical activity levels, and this effect was not found in the overall levels of PA or the reduction of sedentary behavior. The quality assessment revealed that one was good, one was poor and one was fair quality.

## Interventions for both children and adolescents aged 6 to 18 years

Out of the twenty-three reviews, thirteen reviews [15–17, 28–37] considered interventions conducted in both primary and secondary schools for children and adolescents aged between 6 and 18 years.

**Active transport.** Three reviews [29, 32, 34] investigated interventions concerning "active school transport" and "active commuting to and from school". In reviews of [29, 34], acceler-ometers were used to measure PA in all included studies. Larouche et al reported that 13 inter-ventions resulted in a statistically significant increase in active school transport whereas eight reported no changes at all, four included studies recorded inconsistent results and five did not record inferential statistics. Campos-Garzón et al included 14 studies of which 11 were cross-sectional and two longitudinal studies, and their meta-analysis found that the overall active commuting to and from school weighted light physical activity (ACS) was 19.55 min (95% CI: 3.84–35.26; I2 = 99.9%, p < 0.001) and 68.74 min (95% CI: 6.09–131.39; z = 2.15, p = 0.030) during the home-school and school-home trips, respectively. For MVPA, the overall ACS weighted MVPA was 8.98 min (95% CI: 5.33–12.62; I2 = 99.95%, p < 0.001) during the home-school trip and 20.07 min (95% CI: 13.62–26.53; I2 = 99.62%, p < 0.001) during the school-home trip. Schönbach et al included nine studies that investigated seven unique inter-ventions and they reported that only one included study reported positive intervention effects on total MVPA (β = 21.6 [CI95: 8.7, 34.6]), MVPA from cycling (β = 23.0 [CI95: 10.7, 35.4]) and MVPA before/after school (β = 12.8 [CI95: 8.5, 17.2]).

It should be noted that most of the included studies in the reviews of [29, 32] were rated as poor quality and had severe limitations in the design and implementation of interventions, therefore, they attributed a low grade for the overall quality of evidence. For the review con-ducted by [34] most studies had a low risk of bias and medium quality.

**Multicomponent interventions.** In two reviews [16, 17] most of their included interven-tions were concerning diet, education, and PA. The review of [17] was in Latin America and the review of [16] was in the Middle East and Arabic-speaking countries. [17] reviewed 16 interventions most of which targeted diet alone and two focused on PA compared a curricu-lum of PA programming with a conventional PE class, the duration ranged from four months to two years with a minimum follow-up of five months. They reported that above 60% of the included studies recorded a significant effect on at least one dietary/PA or obesity-related out-come, and five included studies reported a significant effect on physical condition tests and/or PA time. While [16] reviewed 17 interventions most of them were multi-component (lifestyle, diet, education) and few studies focused on PA. The duration ranged from six weeks to three years with a maximum follow-up of one year. They reported that eleven included studies recorded statistically significant improvements in the levels of PA among their participants based largely on self-reported outcomes. They also reported that the studies with a follow-up period greater than three months reported sustained PA levels. It should be noted that most of the included studies (13 out of 16) in the review conducted [17] were weak quality. In the review of [16], 14 out of 17 studies had a low risk of bias.

**Single component interventions.** In the review conducted by [28], they aimed to examine the possibility of school-based interventions for promoting PA and PF and preventing obesity. They reviewed 19 studies and most of the interventions were focused on physical activity-related interventions including expanding the duration of PE or altering its content, perform-ing extracurricular physical activities, participating in activities during breaks and lunch breaks, or providing activity breaks in classes other than PE. They recorded that when the BMI variable was examined in studies more focused on PA, the success rate was 72.72% in eight out of 11 studies included and the success rate was found in only two included studies used PA as

support. These findings refer to that PA-oriented interventions are more likely to be successful in the BMI variable. Furthermore, when the PA variable was examined, they found that PA levels of school children significantly increased in a proportion of the studies, and this success rate was achieved in studies focused on orientation and training. This result might show the significance of applying physical activity strategies as the focus of the intervention program to increase the levels of PA [28]. It should be noted that 10 included studies were rated as strong quality whereas nine studies were moderate quality.

**Educational interventions.** The review and meta-analysis conducted by [30] aimed to evaluate the effects of educational interventions on promoting regular PA. They reviewed 14 studies and 8 of them were performed as a controlled clinical trial. They reported that the overall estimate of the mean difference demonstrated that interventions had a statistically significant effect on weight loss, the intervention group had a weight loss of 1.02 kg (95% CI: 0.22–4.79 kg) compared with the control group and the effectiveness in included studies that used combined intervention for a longer period of time was higher than one-dimensional interventions. The quality assessment revealed that six included studies were high quality, four studies were medium and four were weak quality [30].

**Active desks.** The review of [33], investigated the effects of active desks on sedentary behavior, PA, academic achievements, and overall health. They included 23 studies and active desks were in three forms: upright active desk, cycling desk and stability ball. One included study out of 23 found a significant difference in BMI for the interventional group compared to the control group after two years of intervention (−5.24 for BMI percentile) and other studies did not report any change. For sedentary behavior, two included studies noticed that when children used upright active desks spent significantly less time sedentary rather than children in the control group and six included studies reported a significant decrease in sitting time during school hours in the intervention group. Regarding PA, included studies recorded several different outcomes such as light PA, MVPA, step counts, stepping, standing, and walking time. For MVPA, two studies reported contradictory results and one study observed a significant increase in MVPA between pre- and post-intervention. Furthermore, statistically significant increases were reported for standing time in nine included studies. For step counts, one study recorded an increase but without statistical analyses. For stepping time, one study reported an increase in light PA and MVPA with an objective measurement. Regarding academic performance, only two studies that used upright active desks recorded an increase in the intervention group without any change in concentration. The overall quality of the included studies in the review of [33] was low.

**Motivational interventions.** The review of [31] examined motivational interventions based on PA as the precursor of psychosocial benefits among children and adolescents. 45 studies were included, and 39 studies presented a quasi-experimental design and between five and twenty weeks were the average durations of the intervention programs. Vaquero-Solís reported that 23 studies out of 45 demonstrated psychological effects, 10 included studies demonstrated psychosocial effects after the intervention and the rest studies reported changes that were not significant. Moreover, the majority of reviewed studies were based on the theoretical models: Achievement Goal Theory and Self-Determination Theory. Therefore, their review demonstrated the significance of motivational processes for the implementation of PA and sport as a precursor of psychosocial changes highlighting the significance of strategies and the duration of interventions/programmes—ie the longer interventions also have a longer lasting impact on behaviour to maintain significant changes over time. The majority of included studies in review of [31] were rated as high quality with few being medium quality.

## Geographically specific reviews

The review of [15] examined the effectiveness of interventions to promote PA among children and adolescents in Asian countries. They reviewed 30 studies, 21 included studies were school-based, and 16 studies were cluster RCT. Interventions ranged in duration from one day to 60 months (median 4.5 months). Their primary outcome was changed in PA and PA was assessed by device-based tools such as an accelerometer and pedometer in four studies, and the rest used self-reported such as questionnaires. 17 included studies out of 30 used a theoretical framework/ model and eight studies used multiple intervention strategies including a combination of PA sessions, exercise, physical activities such as indoor-outdoor PA, aerobic dance, rope skipping activities and PE sessions. The other included studies used educational interventions. 15 studies showed significant increases in PA behaviour or PF and these studies were of moderate-to-high quality. Their evidence also showed that single or multicomponent interventions for short-term (up to six months) including PA sessions, PE, and health education might increase overall PA in Asian children and adolescents.

## Rural and urban settings

The systematic review and meta-analysis of [36] aimed to evaluate and compare the effect of rural and urban/suburban school-based PA interventions on total PA among youth. They included 33 studies, and the most common intervention component was PE as used 20 times and classroom-based interventions were used 16 times. In rural studies, the meta-analysis indicated that no statistical significance pooled intervention effect on total PA. In subgroup analysis, they found that the magnitude of the effect was consistently small. The test of group differences was significant only when considering studies separated by design (RCT vs. non-RCT), $Q_b$ (38) = 4.97, $p = 0.03$, indicating a significant difference in effect size between interventions with an RCT design and interventions with a non-RCT design. Therefore, in brief, their findings demonstrated that the pooled effect of all school-based interventions to increase total PA was statistically significant but small. When they analysed urban and suburban studies separately, the effect was also statistically significant but small. When they analysed rural studies separately, findings demonstrated a null effect of school-based interventions to increase total PA. It should be noted that the quality of included studies is unknown since [36] did not assess the risk of bias for included studies.

**Impact on MVPA and fitness.** The review of [37], investigated intervention effectiveness to increase MVPA and improve fitness. There were 89 included studies in the review representing 66,752 individuals. They reviewed 40 multi-component interventions, 19 school time PA, 15 enhanced PE, and 14 before and after school programs. The duration of interventions varied greatly from a minimum of 12 weeks to 6 years, with 10 studies reporting intervention periods of 3 years or longer. Their review showed that interventions based in schools may result in little to no increase in time engaged in MVPA (mean difference 0.73 minutes/d, 95% confidence interval 0.16 to 1.30, little to no decrease in sedentary time (MD -3.78 minutes/d, 95% CI -7.80 to 0.24, may improve PF recorded as maximal oxygen uptake (VO2max) (MD 1.19 mL/kg/min, 95% CI 0.57 to 1.82 and may result in a minimal decrease in BMI z-scores (MD -0.06, 95% CI -0.09 to -0.02. The overall quality of the included studies in the review of [37] was as follows: 33 studies were moderate quality, and 56 studies were low quality.

**Impact on HRQoL.** The meta-analysis conducted by [35], aimed to assess the effects of PA interventions on several domains of health-related quality of life (HRQoL) and to examine the effectiveness of interventions on the same domains. They reviewed 17 different interventions and most of them were adding extra (PE) lessons to the regular school curriculum with a few of them being short active breaks. Their meta-analysis indicated that pooled effect size

(95% CI) estimates for the effect of PA on HRQoL were as follows: 0.179 (0.045, 0.002) for total HRQoL score, 0.192 (0.077, 0.306) for physical well-being, 0.158 (0.080, 0.237) for psychological well-being. In brief, their results conclude that physical exercise improved overall HRQoL and several HRQoL domains, such as physical well-being, psychological well-being, autonomy and parent relation, and social support and peers. However, their findings indicate no positive effect of the exercise programs on the school environment domain. The risk of bias for the included studies in the review of [35] showed that six out of 17 were at high risk of bias, four studies were medium, and seven studies were at low risk of bias.

## Quality assessment of included reviews

The methodological quality of the included reviews was classified into four categories: high, moderate, low, or critically low overall quality. Seven reviews were of high quality [4, 5, 14–16, 22, 23], four reviews were of moderate quality [17–20], five reviews were of low quality [6, 8, 9, 12, 21], and seven reviews were critically low [1–3, 7, 10, 11, 13]. Most reviews did not report on the sources of funding for the studies included, except one review [37]. Also, most studies did not provide a list of excluded studies nor justify the exclusions, except for two reviews [16, 23]. Three out of the twenty-three reviews failed to assess the risk of bias in the included studies, and two of them did not consider the risk of bias in individual studies when interpreting/discussing the results [20, 24, 36]. Furthermore, seven studies did not explain their selection of study designs for inclusion in their reviews [16–19, 28, 29, 31]. Nine studies omitted the components of PICO in their research questions and inclusion criteria [17–20, 25, 26, 28, 33, 35], and the authors of seven studies did not perform duplication in study selection or data extraction [16, 17, 19, 20, 22, 31, 36].

Eight out of the twenty-three reviews conducted a meta-analysis. Four of these studies used partially appropriate methods for the statistical combination of results and assessed the potential impact of the risk of bias in the included studies on the results and evidence synthesis [19, 22, 34, 36]. The findings of the quality assessment are presented in S4 Appendix in S1 File).

## Discussion

The main purpose of this umbrella review was to comprehensively evaluate the effectiveness of school-based physical activity interventions among children and young people aged 6 to 18 years internationally. A total of 23 reviews were included, with seven reviews implemented in primary schools among children aged 6 to 11 years, three reviews implemented in secondary schools among adolescents aged 12 to 18 years, and thirteen reviews targeted both children and adolescents aged 6 to 18 years. Our ability to come to solid conclusions was limited because the quality of the systematic reviews or meta-analyses was low, or because, as mentioned in several reviews, the quality of the included studies was very weak. Nevertheless, a few approaches seem more hopeful in encouraging more physical activity than others.

This umbrella review identified various school-based interventions, presenting mixed findings, with effectiveness appearing to vary by age. Among the reviews focusing on interventions in primary schools for children aged 6 to 12 years, two interventions—the Daily Mile Programme [20, 23] and Active Break intervention [25]—were highlighted as promising for increasing PA, demonstrating significant improvements in physical activity, physical fitness, and MVPA. The Daily Mile Programme, implemented in over 15,600 schools worldwide, has shown rapid adoption over the last decade [23], with evidence indicating that long-term participation can enhance MVPA by nearly ten minutes a day and positively impact physical fitness. However, children participating in The Daily Mile programme do not maintain MVPA intensity throughout the entire 15 minutes, though any increase in MVPA is beneficial for

cardiometabolic health [38]. Nevertheless, to accurately assess the sustainability and full impact of The Daily Mile, higher-quality research with proper randomisation and extended follow-up periods is recommended [23].

Active break interventions were performed in three forms: active breaks, curriculum-focused active breaks, and physically active lessons. Two reviews recorded a significant increase in PA and MVPA with no significant reduction in sedentary behaviour [21, 25]. Therefore, these two specific interventions (TDM and Active breaks), seem to contribute to achieving the public health recommendation of 60 minutes of MVPA per day for children.

Among the reviews that investigated interventions conducted in secondary schools for adolescents aged 12 to 18 years. Multicomponent interventions in two reviews [18, 19] demonstrated a very small and nonsignificant effect on PA levels. These findings are in line with the previous umbrella review in the same field [9] as they reported that school-based interventions targeting adolescents do not have significant effects and are of questionable value. Owen et al. suggest that during adolescence, complex physiological and psychological changes occur, which may complicate efforts to change behavior [19].

Reviews including both children and adolescents found mixed findings for school-based interventions. Active transport interventions demonstrated a statistically significant increase in active school transport when PA was objectively measured [29]. For MVPA, the overall ACS weighted MVPA was 8.98 min (95% CI: 5.33–12.62; I2 = 99.95%, p < 0.001) during the home-school trip and 20.07 min (95% CI: 13.62–26.53; I2 = 99.62%, p < 0.001) during the school-home trip (Campos-Garzón et al., 2023). Therefore, active transport interventions may contribute 50% of the physical activity recommendations in school aged children on school days when both trip directions were actively performed.

Single component interventions with a focus on PA-related interventions including expanding the duration of PE or altering its content, performing extracurricular physical activities, participating in activities during breaks and lunch breaks, or providing activity breaks in classes other than PE demonstrated a small but significant increase in PA levels especially when they focused on orientation and training [28]. Educational interventions that concerning diet, PA and education showed a statistically significant effect on weight loss and a significant increase in PA especially when physical activity programs employed a multi-component approach that integrated into the school curriculum, and included teachers, parents, and students [30]. These results are in line with previous reviews indicating that programs that combine diet and PA produced greater benefits than single strategies alone [39–41]. Furthermore, we found that a long-term intervention was another key component of effective interventions. The results of this review are consistent with those of previous studies that found that school-based interventions became successful when they had a duration of one school year or more and less successful in those that lasted less than six months [6].

Active desk interventions showed a small effect on BMI reduction, a significant decrease in sitting time during school hours, and statistically significant increases in standing time [33]. Motivational interventions showed psychosocial effects after the intervention when they were based on theoretical models such as achievement goal theory and self-determination theory [31]. These results in line with the previous narrative review [6] examined the effectiveness of school-based interventions in Europe and indicated that ineffective school-based interventions were not based on a well-defined theoretical framework. This indicates the importance of theoretical models in designing school-based PA interventions.

Multi-component interventions demonstrated little to no increase in MVPA, little to no decrease in sedentary time, a minimal decrease in BMI and little improve in PF [37]. Finally, adding extra (PE) lessons to the regular school curriculum with a few of them being short active breaks showed positive effect on several domains of health-related quality of life such as

physical well-being and psychological well-being [35]. Nevertheless, schools remain an important setting for promoting PA, since they interact more frequently with school-aged children during the first two decades of their lives than any other institution. Furthermore, as indicated by [42] youth's ability to increase their level of physical activity and sustain it may be heavily influenced by their social environment and their social support system. It is unlikely that someone will begin to walk to school if no one walks to school [42]. Additionally, [31] reported in their review that the behaviour of peers has a profound effect on young people's behaviour.

Both the methods and reporting of systematic reviews in the topic area could be improved. For example, most reviews failed to report on the sources of funding for the studies included in the review. Also, most studies failed to provide a list of excluded studies and justify the exclusions. Three reviews out of 23 reviews failed to assess the risk of bias in included studies and two of them did not account for the risk of bias in individual studies when interpreting/ discussing the results [20, 24, 36]. Nine studies did not include the components of PICO in the research questions and inclusion criteria. Despite this, it was difficult to draw concrete conclusions in light of the low quality of some included studies. Consequently, further high-quality research is needed to determine the effectiveness of physical activity interventions.

## Strengths and limitations of primary studies included in reviews

The key strength of primary studies included in reviews is the range and diversity of programmes evaluated and the range of different outcome measures. In addition, eleven reviews included high to medium-quality primary studies. However, there were some limitations: firstly, the quality of primary studies in some reviews cited as a limitation by review authors. Further, some authors indicated that the primary studies included had a high risk of bias and were unclear in some cases [17, 25, 29, 32, 33, 35, 37]. So, it is impossible to fully trust the conclusions of a systematic review if the primary studies were flawed. Secondly, related to the trade-off between evaluating "real world" programmes and the lack of rigorous RCTs—and the impossibility of blinding which means that depending on self-reported outcomes was problematic.

## Strengths and limitations of the umbrella review

The key strength of this umbrella review is that it is the first to systematically assess the most recent scientific evidence on school-based physical activity interventions to promote physical activity among children and adolescents around the world. Furthermore, an in-depth quality assessment was conducted using the latest version of the AMSTAR 2, a tool for assessing systematic reviews. In addition, bringing these findings together has provided a comprehensive picture of the school-based physical activity interventions that are most likely to increase school children's and adolescents' levels of physical activity. However, this umbrella systematic review has some limitations. Firstly, the searches were limited to studies published in English and Arabic language which may led to missing potentially relevant studies. Secondly, with consideration of the heterogeneity in primary studies designs in each review, different types of interventions, outcomes, duration and measures restricted to the potential synthesis of the findings from different reviews and to directly compare the effectiveness of different intervention types. Thirdly, Our umbrella review includes 23 systematic reviews, which collectively include 691 primary studies. The scope and volume of these studies make it logistically difficult to accurately track and report each study's country of origin within the confines of our current review format. Given the extensive nature of the data and our resource constraints, we categorised the reviews by continent to provide a manageable and still informative geographical overview. This approach was intended to balance detail with clarity, allowing us to discuss regional

trends without becoming overly granular, which could potentially obscure the broader patterns relevant to our findings.

## Implications of the findings for Policy Makers and Schools

The findings of this umbrella review have shown that some interventions were promising in increasing physical activity levels in school children and adolescents such as the Daily Mile programme, active breaks, active desks, and multi-component interventions. Other interventions such as multi-component interventions concerned with PA, nutrition and education shown a potential to increase PA and decrease BMI. While effective programmes have been identified, their applicability may vary globally. Thus, future research should aim to enhance the quality of evidence and broaden the implementation context. Moreover, there is a need for more studies with robust designs, like rigorous randomised controlled trials, that have adequate sample sizes, proper control groups, sufficient follow-up periods extending well beyond the intervention, and that employ validated measures of PA and assess sustainability issues.

## Conclusion

This umbrella review evaluated the literature on the effectiveness of school-based physical activity interventions for children and young people aged 6 to 18 years internationally. The Daily Mile, Active Break, and Active Transport interventions showed promise in enhancing PA levels, while multi-component interventions were effective in reducing BMI. Long-term interventions, supported by rigorous theoretical and methodological frameworks, also appeared to increase PA levels.

The findings from this review are crucial for shaping future research directions, emphasising the need for enhanced methodological quality and more robust RCTs. These should include adequate sample sizes, appropriate control groups, and extended follow-up periods to accurately assess the sustainability and true effectiveness of interventions. Moreover, the review underscores the importance of integrating validated physical activity measures into future studies, alongside an evaluation of sustainability issues.

## Supporting information

**S1 Checklist. PRISMA 2020 checklist.**
(DOCX)

**S1 File.**
(DOCX)

## Author Contributions

**Conceptualization:** Abdullah Alalawi, Lindsay Blank.

**Data curation:** Abdullah Alalawi.

**Supervision:** Lindsay Blank, Elizabeth Goyder.

**Validation:** Lindsay Blank.

**Writing – original draft:** Abdullah Alalawi.

**Writing – review & editing:** Lindsay Blank, Elizabeth Goyder.

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
