## [Decision Letter · Decision Letter 0]

19 Apr 2024

PONE-D-24-10547The Effectiveness of School-Based Interventions to Increase Physical Activity among Children and Young People aged 6 to 18 years: A Systematic Review of ReviewsPLOS ONE

Dear Dr. Alalawi,

Thank you for submitting your manuscript to PLOS ONE. After careful consideration, we feel that it has merit but does not fully meet PLOS ONE’s publication criteria as it currently stands. Therefore, we invite you to submit a revised version of the manuscript that addresses the points raised during the review process.

We look forward to receiving your revised manuscript.

Kind regards,

Henri Tilga, PhD

Academic Editor

PLOS ONE

Journal Requirements:

3. We note you have included a table to which you do not refer in the text of your manuscript. Please ensure that you refer to Table 2 in your text; if accepted, production will need this reference to link the reader to the Table.

Reviewers' comments:

Reviewer's Responses to Questions

**Comments to the Author**

1. Is the manuscript technically sound, and do the data support the conclusions?

Reviewer #1: Yes

Reviewer #2: Yes

2. Has the statistical analysis been performed appropriately and rigorously? 

Reviewer #1: I Don't Know

Reviewer #2: Yes

3. Have the authors made all data underlying the findings in their manuscript fully available?

Reviewer #1: Yes

Reviewer #2: Yes

4. Is the manuscript presented in an intelligible fashion and written in standard English?

Reviewer #1: Yes

Reviewer #2: Yes

5. Review Comments to the Author

Reviewer #1: Title

1. Very long title, try to summarize the title

Keywords

2. The words "systematic review", "school-based", "physical activity" and "interventions" are already included in the title, here they should be replaced by synonyms.

Introduction

3. In the introduction you mention in particular obesity and sedentary lifestyle, and talk briefly about physical inactivity, I believe it would be important to reinforce in the introduction with quantitative data the size (percentage) of this physical inactivity among this world population. I believe it is important since all the interventions studied certainly aimed to reduce sedentary behavior and/or increase the level of physical activity and consequently remove this population from the "physically inactive" classification.

4. About PRISMA item 3 "Describe the rationale for the review in the context of existing knowledge" it needs to be improved.

Methods

5. Inclusion and Exclusion Criteria: The exclusion criteria serves to exclude studies that were included according to my inclusion criteria, but which, due to some specific characteristic, do not meet the other inclusion criteria. So when you say that the inclusion criterion is "Interventions that were implemented in schools and quantitative evaluations of PA" you already make it clear that " Reviews included interventions not based on schools" will not be included, so I suggest rethinking all the exclusion criteria, as most of the exclusion criteria seem to me to already be included in the inclusion criteria.

6. The reference [9] that you mention as if your study were a "continuation" used the following databases PubMed, Scopus and the Cochrane Library. So I would like to know what the justification is for using CINAHL and not the Cochrane Library? Considering the characteristics of the databases, I believe that the Cochrane Library would be more appropriate. Another detail is that the reference you cited used 3 languages (English, Italian and Spanish) and you only used two (Arabic and English) I believe it would be important to consider at least one more language.

Results

7. Appendix 2. Table 2: Characteristics of the included reviews. In the first column it would be very important to include in which country the studies were developed, so that you could also summarize by country in the item "Review characteristics" and not just by continent as stated.

8. About Appendix 2. Table 2: Characteristics of the included reviews. It would be very important for it to appear in the body of the text and not just as an appendix, but for that to happen it should be "summarized" as it is very extensive, I believe that some "column" information could be excluded without losing the essence.

9. In this excerpt "Quality Assessment of Included Reviews The methodological quality of the included reviews was classified into four categories: high, moderate, low, or critically low overall quality. Seven reviews were of high quality, four reviews were of moderate quality, five reviews were of low quality, and seven reviews were critically low." It is important to include the references of the 7 studies that have high quality, the 4 moderate quality and so on, this information must be verified throughout the results section.

Conclusion

10. better explain the contribution of this review to future studies

Reviewer #2: attached

6. PLOS authors have the option to publish the peer review history of their article (what does this mean?). If published, this will include your full peer review and any attached files.

Reviewer #1: **Yes: **REGINA MÁRCIA FERREIRA SILVA

Reviewer #2: No

---

## [Author Response · Author response to Decision Letter 0]

22 Apr 2024

Dear reviewers

Thank you for reviewing our paper for publication in PLOS One. Please see document "Response to Reviewers". We have addressed your valuable suggestions and comments.

---

## [Decision Letter · Decision Letter 1]

14 May 2024

Umbrella Review of International Evidence for the Effectiveness of School-based Physical Activity Interventions

PONE-D-24-10547R1

Dear Dr. Alalawi,

We’re pleased to inform you that your manuscript has been judged scientifically suitable for publication and will be formally accepted for publication once it meets all outstanding technical requirements.

Kind regards,

Henri Tilga, PhD

Academic Editor

PLOS ONE

Additional Editor Comments (optional):

Authors have done well job on revising the manuscript.

Reviewers' comments:

Reviewer's Responses to Questions

**Comments to the Author**

1. If the authors have adequately addressed your comments raised in a previous round of review and you feel that this manuscript is now acceptable for publication, you may indicate that here to bypass the “Comments to the Author” section, enter your conflict of interest statement in the “Confidential to Editor” section, and submit your "Accept" recommendation.

Reviewer #2: (No Response)

2. Is the manuscript technically sound, and do the data support the conclusions?

Reviewer #2: (No Response)

3. Has the statistical analysis been performed appropriately and rigorously? 

Reviewer #2: (No Response)

4. Have the authors made all data underlying the findings in their manuscript fully available?

Reviewer #2: (No Response)

5. Is the manuscript presented in an intelligible fashion and written in standard English?

Reviewer #2: (No Response)

6. Review Comments to the Author

Reviewer #2: (No Response)

7. PLOS authors have the option to publish the peer review history of their article (what does this mean?). If published, this will include your full peer review and any attached files.

Reviewer #2: No

---

## [Editor Report · Acceptance letter]

21 May 2024

PONE-D-24-10547R1 

PLOS ONE

Dear Dr. Alalawi, 

I'm pleased to inform you that your manuscript has been deemed suitable for publication in PLOS ONE. Congratulations! Your manuscript is now being handed over to our production team.

Kind regards, 

on behalf of

Dr. Henri Tilga 

Academic Editor

PLOS ONE